# Antibacterial, Mutagenic Properties and Chemical Characterisation of Sugar Bush (*Protea caffra* Meisn.): A South African Native Shrub Species

**DOI:** 10.3390/plants9101331

**Published:** 2020-10-09

**Authors:** McMaster Vambe, Adeyemi O. Aremu, Jude C. Chukwujekwu, Jiri Gruz, Andrea Luterová, Jeffrey F. Finnie, Johannes Van Staden

**Affiliations:** 1Research Centre for Plant Growth and Development, School of Life Sciences, University of KwaZulu-Natal, Pietermaritzburg, Private Bag X01, Scottsville 3209, South Africa; Vambem@ukzn.ac.za (M.V.); Chukwujekwu@ukzn.ac.za (J.C.C.); Finnie@ukzn.ac.za (J.F.F.); 2Indigenous Knowledge Systems Centre, Faculty of Natural and Agricultural Sciences, North-West University, Private Bag X2046, Mmabatho 2745, South Africa; 3Laboratory of Growth Regulators, The Czech Academy of Sciences, Institute of Experimental Botany and Palacký University, Šlechtitelů 27, CZ-78371 Olomouc, Czech Republic; jiri.gruz@upol.cz (J.G.); andrea.luterova@upol.cz (A.L.)

**Keywords:** antimicrobial, drug-resistance, diarrhoea, GC-MS, mutagenicity, Proteaceae, UHPLC-MS/MS

## Abstract

*Protea caffra* is used as a diarrhoeal remedy in South African herbal medicine, however, its pharmacological properties remain largely unknown. In the present study, extracts from different *Protea caffra* organs were screened against drug-sensitive and -resistant diarrhoeagenic pathogens using the microdilution assay (minimum inhibitory concentration, MIC). Twig extracts (70% methanol, MeOH) of the plant were purified and the resultant fractions screened for antibacterial properties (MIC). The chemical profiles of the fractions were determined by Gas Chromatography-Mass Spectrometry (GC-MS), while ultra-high-performance liquid chromatography-tandem mass spectrometry (UHPLC-MS/MS) was used to quantify the phenolic acids in the plant. The mutagenic properties of bioactive extracts were assessed using the Ames test. The extracts demonstrated weak-moderate antibacterial properties (MIC: 0.3–0.6 mg/mL). A cold ethyl acetate fraction of MeOH twig extract exhibited significant antibacterial properties (MIC = 0.078 mg/mL) against *Enterococcus faecalis*. The presence of antibacterial compounds (1-adamantane carboxylic acid, heptacosanol, levoglucosan, nonadecanol) in the plant was putatively confirmed based on GC-MS analysis. Furthermore, UHPLC-MS/MS analysis revealed varying concentrations of phenolic acids (0.08–374.55 µg/g DW). Based on the Ames test, the extracts were non-mutagenic thereby suggesting their safety. To a certain degree, the current study supports the traditional use of *Protea caffra* to manage diarrhoea among local communities in South Africa.

## 1. Introduction

Diarrhoea is an intestinal disorder characterised by the passage of frequent loose or watery stools in a period of 24 h [1]. Infectious diarrhoea remains one of the leading causes of child mortality worldwide. In 2015 alone, diarrhoeal diseases caused an estimated 1.3 million deaths worldwide and were the fourth leading causes of death among children under the age of 5 years [2]. Some of the prominent etiological agents of community and hospital-aquired diarrhoea include *Escherichia coli*, *Enterococcus faecalis*, *Klebsiella pneumonia* and *Staphylococcus aureus* [3,4]. Globally, medicinal plants are used to manage diarrhoeal symptoms [5]. *Acacia nilotica* (seed powder), *Bacopa monnieri* (leaf decoction), *Rheum palmatum* (rhizome infusions), *Santalum album* (rhizome infusion) are, for instance, listed in the Asian Pharmacopeia as traditional herbs with anti-diarrhoeal properties [6,7,8]. In Europe, diarrhoea is managed traditionally using plants such as *Matricaria chamonilla* (dry flowering decoctions), *Solanum tuberosum* (tuber decoctions) and *Vaccinium myrtillus* (fruit decoctions) [9]. African herbs commonly used to treat the disease include *Elephantorrhiza elephantina* (root decoctions), *Euphorbia hirta* (leaf macerate), *Heinsia pulchella* (root bark decoction), *Ozoroa insignis* (bark decoctions), *Psidium guajava* (leaf infusions), *Sclerocarya birrea* (leaf and bark infusions), *Solanum supinum* (root decoction), *Terminalia sericea* (root infusions) and *Ximenia caffra* (roof decoctions) [10,11,12,13]. In South Africa, *Protea caffra* Meisn (Proteaceae) is among the common plants utilised by local herbalists for treating diarrhoea [14]. 

*Protea caffra* is a dicotyledonous shrub (3 m) that grows in different parts of Mozambique, South Africa and Zimbabwe [15,16,17]. The plant is native to South Africa where it is locally known as the common sugarbush, Natal sugarbush (English), *gewone suikerbos*, *waboom* (Afrikaans), *isadlunge*, *indlunge*, *isiqwanwe* (Isixhosa), *uhlinkihane* (isiZulu), *tshididiri, tshidzungu* (TshiVhenda), *mahlako*, *mogalagala*, *segwapi*, *sekila* and *tshidzungu* (Sotho) [18,19,20]. As applicable with other members of the genus *Protea*, it has characteristic large beautiful flower heads which have made it an important ornamental plant in southern Africa and other parts of the world [17]. Apart from having important horticultural purposes, *Protea caffra* has several ethnomedicinal applications. For instance, the aqueous infusions of the root and stem barkare used to manage bleeding stomach ulcers, diarrhoea or as enemas [14,20]. According to Zukulu [21], the roots of the plant are used to prepare *umhlabelo*, a decoction used to help heal broken bones. The fruit and stem bark are also used to manage dizziness, while decoctions of dried seeds are used to manage different psychological disorders [22]. A study by Semenya et al. [23], revealed that the Bapedi traditional healers of South Africa use *Protea caffra* seed infusions to manage chlamydia, a sexually transmitted bacterial infection. However, the scientific basis for the majority of its traditional uses are currently lacking or limited. In addition, the phytochemical and toxicological evaluations of *Protea caffra* remain pertinent to contribute toward its wider acceptance. The present study investigated the antibacterial (against drug-sensitive and -resistant strains), mutagenic and phytochemical properties of different parts of *Protea caffra*.

## 2. Results

The antibacterial MIC values of evaluated plant extracts are presented in Table 1. The extracts were classified as having significant (MIC ≤ 0.1 mg/mL), moderate (0.1 < MIC ≤ 0.625 mg/mL) or weak (MIC > 0.625 mg/mL) antibacterial properties [24]. All aqueous extracts yielded MIC values >2.5 mg/mL (Appendix A). Generally, the evaluated plant parts demonstrated moderate antibacterial properties with MIC values ranging from 0.3–0.6 mg/mL. Interestingly, extracts from the seed (MeOH and petroleum ether) and twigs (MeOH) demonstrated noteworthy antibacterial activities against Gram-negative bacterial strains (*Escherichia coli* and *Klebsiella pneumonia*). It was also worth noting that the methanolic leaf and twig extracts of the plant demonstrated promising bacteriostatic properties (MIC = 0.63 mg/mL) against the penicillin-resistant *Staphylococcus aureus*. However, the other drug-resistant bacterial strains were not susceptible to the plant extracts (MIC > 2.5 mg/mL) and hence were excluded from Table 1. 

The MeOH twig extract demonstrated extended-spectrum antibacterial properties (Table 1), an observation that stimulated interest in determining its phytochemical profile. Acetone, cold ethyl acetate and hot ethyl acetate were used to partition the compounds in the biologically active extract and the resultant fractions were screened for antibacterial properties [25]. As shown in Table 2, the acetone and methanol sub-fractions demonstrated very weak antibacterial properties (MIC ≥ 2.5 mg/mL). The best antibacterial property was however, exhibited by the cold ethyl acetate sub-fraction which was active against both Gram-negative (*Escherichia coli, Klebsiella pneumoniae*) and Gram-positive (*Enterococcus faecalis*, *Staphylococcus aureus*) bacterial strains (MIC range: 0.078–0.6 mg/mL). The hot ethyl acetate sub-fraction demonstrated moderate activities against *Enterococcus faecalis* (0.3 mg/mL) and *Staphylococcus aureus* (0.6 mg/mL). However, none of the fractions evaluated demonstrated noteworthy antibacterial activities against drug-resistant bacterial strains (Table 2).

GC-MS data analysis revealed that the aqueous (70%) MeOH extract of *Protea caffra* twigs consisted of 15 compounds (Table 3, Appendix A). No peaks were obtained from the methanol sub-fraction. The major phytocompounds in the cold ethyl acetate sub-fraction were polygalitol (34.76%), phenol, 4-(1,1,3,3-tetramethylbutyl) (9.8%), Spiro-1-(cyclohex-2-ene)-2’-(5’-oxabicyclol) (8.2%), 1-adamantane carboxylic acid (8.07%) and carbamic acid (7.03%), which together accounted for approximately 60% of the compounds found in the sub-fraction (Table 3). The hot ethyl acetate sub-fraction consisted of 1-heptacosanol (70.57%) as its major component (Table 3). Oxalyl chloride (51.12%) and polygalitol (48.88%) were the only compounds detected in the acetone sub-fraction.

The UHPLC-MS/MS analysis revealed that *Protea caffra* contained varying quantities of both hydroxybenzoic and hydroxycinnamic acids (Table 4 and Table 5). Overall, the most abundant hydroxybenzoic (*p*-hydroxybenzoic acid, 374.55 µg/g DW) and hydroxycinnamic (caffeic acid, 266.37 µg/g DW) acids were detected in the leaves. The leaves also contained the least abundant hydroxycinnamic acids (sinapic acid, 0.08 µg/g DW), while the bark contained the least abundant hydroxybenzoic acid (salicylic acid, 0.1 µg/g DW). 

None of the evaluated plant extracts demonstrated concentration-dependent increase in the number of His^+^ revertants (Table 6). The average TA98 revertants ranged from 6.4–29.0, while the TA102 revertants ranged from 109.7–284.3. The corresponding average number of TA98 and TA102 revertants in the negative control were 19.1 and 145.2, respectively. The evaluated extracts were therefore non-mutagenic against TA98 and TA102 tester strains [26].

## 3. Discussion

Given that water is one of the most commonly used solvent in folklore medicine [27], it was included in the present study to mimic traditionally prepared herbal medicines. However, water extracts have been widely reported to exhibit poor antibacterial activity. This is attributed to the fact that many antibacterial phyto-compounds are non-polar or have intermediate polarity and as such cannot be readily extracted from plant material using water [28,29]. In the current study, it is possible that the antibacterial compounds in *Protea caffra* water extracts occurred in very low, sub-lethal concentrations resulting in poor antibacterial activity (Appendix A). It should, however, be noted that some phyto-compounds indirectly help patients fend off pathogenic infections by acting as immune boosters [30]. Furthermore, some of the bioactive compounds in water extracts may exist as pro-drugs, which only become bactericidal once they are modified in the human body [27]. The presence of immune stimulators and pro-drugs in plant extracts cannot be detected using the techniques employed in the current study, as such, further studies are warranted to determine the chemical profiles of traditionally prepared herbal medicines. 

The current study revealed for the first time that *Protea caffra* has potent and extended-spectrum antibacterial properties, which could be attributed to a wide range of biologically active compounds present in the plant. The twigs, for instance, contained 8 putative antibacterial compounds (caffeic, *p*-coumaric, gallic, ferulic chlorogenic acids, adamantyl heterocycle, heptacosanol and nonadecanol, Table 3, Table 4 and Table 5) and they demonstrated moderate antibacterial properties against the majority of the evaluated bacterial strains (Table 1). In particular, they also exhibited significant antibacterial activities against *Enterococcus faecalis* (MIC = 0.078 mg/mL, Table 2). The antibacterial properties observed in the present study were generally comparable to those previously reported for some South African medicinal plants such as *Newtonia hildebrandtii*, *Newtonia buchanannii*, *Ozoroa insignis*, *Syzgium cordatum*, *Terminalia. sericea*, and *Trichilia emetica* (MIC range: 0.1–0.6 mg/mL) [4,12,13,31].

Relatively higher concentrations of caffeic and *p*-coumaric acids (>10 µg/g DW) in the seed (Table 5) could have contributed to the inhibitory effects MeOH seed extracts had on the growth of *Escherichia coli*, *Klebsiella pneumoniae* and *Staphylococcus*
*aureus* (Table 1). The current observations could perhaps explain why some herbalists in South Africa use *Protea caffra* seeds to manage bacterial infections including chlamydia [32]. The presence of different types of antibacterial compounds in the leaves and twigs could also perhaps justify the observed moderate antibacterial properties these organs had against penicillin-resistant *Staphylococcus aureus* (Table 1). 

The antibacterial compounds present in *Protea caffra* worked in different synergistic combinations or individually to affect the observed bacteriostatic properties. Different groups of phenolic compounds often exhibit unique antibacterial mechanisms. Gallic and ferulic acids for example, exert their bactericidal effects by binding to and rupturing bacterial cell membranes [33]. Chlorogenic acid on the other hand, binds to bacterial membranes and increases their permeability. This in turn causes the leakage of both cytoplasmic and nuclei material, eventually leading to bacterial cell death [34]. *P*-coumaric acid also causes intracellular material leakages and interferes with bacterial DNA replication and gene expression [35]. 

Apart from phenolic acids, *Protea caffra* contained additional antibacterial compounds as revealed by GC-MS analysis. For instance, 1-Adamantyl heterocycle, which was detected in the twigs of *Protea caffra,* is usually incorporated into anti-infectious molecules to improve their efficacy [36]. Several potent antimicrobial and antiviral agents such as Rimantadine [37], Oxadiazole [38], Isoxazole [39] and Thiadiazole [40] are all 1-adamantanyl derivatives. It was of great interest to note that *Protea caffra* produces levoglucosan (Table 3), an important source of C1-C10 and C1-C13 carbon skeletons used to produce the antibiotics erythromycin A and B, [41,42]. 1-Heptacosanol, another compound detected in *Protea caffra*, is a fatty alcohol present in plants [43], marine algae [44] and cuttlefish, *Sepiella inermis* [45]. The compound has potent antimicrobial properties [46]. The presence of 1-heptacosanol in *Protea caffra* suggests that the plant might have potent antioxidant [44,47], nematocidal [48] and antidiabetic [49] properties. Unlike 1-nonadecanol which is a known antibacterial phyto-compound [50], polygalitol has not yet been demonstrated to have antibacterial properties, but has been detected in several plant extracts with potent antibacterial activities [51]. Polygalitol was the most abundant compound in the cold ethyl acetate fraction which demonstrated broad-spectrum antibacterial activities in the present study (Table 2). Further studies are warranted to determine the compound’s phytochemical properties. Given that 1-heptacosanol, an antibacterial compound, was the major phyto-chemical constituent in the hot ethyl acetate fraction, it is logical to suggest that it was probably the one that inhibited the growth of both *Enterococcus faecalis* and *Staphylococcus aureus* (Table 2). Polygalitol and other phyto-compounds detected in this sub-fraction could also have contributed to the observed antibacterial activities. Oxalyl chloride detected in the acetone sub-fraction is a synthetic compound used in oxidative processes involved in manufacturing antibiotics, pesticides, herbicides and other organic products [52,53]. There are no indications in the current literature suggesting that the compound is produced by plants and/or that it is biologically active. Oxalyl chloride was therefore probably incorporated into *Protea caffra* tissues from an external source. Further investigation would reveal which of the two compounds present in the sub-fraction (acetone) was responsible for the weak antibacterial activity observed against *S*. *aureus* (MIC ≥ 1.25 mg/mL). The principle antibacterial compounds in the *Protea caffra* should, however, be unequivocally identified and their respective antibacterial mechanisms elucidated.

Given that some plants are inherently toxic [54], the safety of traditional herbal remedies remains a serious cause of concern. It was encouraging to note that all evaluated plant extracts were non-mutagenic against both *Salmonella typhimurium* tester strains (TA98 and TA102, Table 6). Based on accessed literature, none of the plant species within the genus Protea have been reported to have potential toxic effects on humans. It is, however, important to note that while some medicinal plants exhibit non-mutagenic effects in vitro, they may possess cytotoxic effects [55]. It should also be kept in mind that besides mutagenesis, carcinogens can also induce cancerous growth in animals through altering intracellular signals and gene expressions both of which are not detected by the Ames test [56]. It is therefore imperative that further toxicological studies be conducted to ascertain the plant’s safety.

## 4. Materials and Methods 

### 4.1. Plant Material Collection, Sample Preparation and Extraction

Plant samples were collected, prepared and preserved as previously described [57]. The plant was positively identified by the Curator of the Bews Herbarium [University of KwaZulu-Natal (UKZN), Pietermaritzburg, South Africa] and the voucher specimen (NU0048533) deposited in the UKZN Bews Herbarium.

At a ratio of 10:1 (10 mL/g), dry powdered samples were mixed with different solvents (water, methanol = MeOH, dichloromethane = DCM, and petroleum ether = PE) and stirred in a rotary shaker (Edmund Bühler, Tübingen, German) for 12 h at 150 rpm at room temperature, after which they were sonicated for 1 h on ice (Julabo GMBH, Seelbach, Germany). Organic solvent extracts were filtered using Whatman No. 1 filter paper under vacuum and later concentrated using a rotary evaporator (Heldolph vv 2000, Germany) at 35 °C. Concentrated organic solvent extracts were transferred into glass pill vials and air-dried in front of a fan. All water extracts were freeze-dried. The resultant dried extracts were kept in closed glass pill vials in the dark at 10 °C until required for further use. 

### 4.2. Antibacterial Susceptibility Test

Minimum inhibitory concentration (MIC) values of aqueous and organic solvent extracts obtained from the plant were determined using 96-well microplates (Greiner Bio-one, Germany) as previously described [25]. Dried plant extracts were re-suspended in 2% dimethylsulfoxide (DSMO, 10 mg/mL) after which they were serially diluted (2-fold) with sterile distilled water. Neomycin (1 mg/mL) was used as a positive control. Sterile distilled water and 2% DSMO were included as the negative controls for each bacterial strain (*Escherichia coli* ATCC 11775, *Enterococcus faecalis* ATCC 19433, *Klebsiella pneumoniae* ATCC 13883, MDR *E. coli* ATCC 25218, MDR *Klebsiella pneumoniae* ATCC 70603, drug-sensitive *Staphylococcus aureus* ATCC 12600 and penicillin-resistant *Staphylococcus aureus* ATCC 11632). Diluted overnight cultures were used at a final inoculum of ≈ 5 × 10^5^ cfu/mL. 

### 4.3. Liquid-to-Liquid Fractionation and Gas Chromatography-Mass Spectroscopy (GC-MS)

Concentrated aqueous (70%) MeOH extract of *Protea caffra* twigs was sequentially extracted with cold ethyl acetate (10 °C, 3 × 50 mL), hot ethyl acetate (50 °C, 3 × 50 mL) and acetone (room temperature, 3 × 100 mL). The resultant fractions were separately concentrated to dryness *in vacuo* to give four solvent fractions: acetone, cold ethyl acetate, hot ethyl acetate and methanol. The four fractions were screened for antibacterial activities against seven bacterial strains (*Enterococcus faecalis, Escherichia coli,* MDR *Escherichia coli*, *Klebsiella pneumoniae*, MDR *Klebsiella pneumoniae*, *Staphylococcus aureus* and penicillin-resistant *Staphylococcus aureus*) as previously described by Eloff [25].

GC-MS analysis of the four fractions was carried out at the School of Chemical and Physical Sciences, University of KwaZulu-Natal, Pietermaritzburg, South Africa, using a Shimadzu QP-2010 SE Gas Chromatography coupled with (an Agilent) 5973 Mass Selective detector and driven by Agilent Chemstation software. A Zebron ZB-5MSplus capillary column (30 m × 0.25 mm internal diameter, 0.25 µm film thickness) was used. The carrier gas was ultra-pure helium at a flow rate of 1.0 mL/min and a linear velocity of 37 cm/s. Three microlitres of the sample were injected into the column with the injector temperature set at 250 °C. The initial oven temperature was at 60 °C which was programmed to increase to 280 °C at the rate of 10 °C per min with a hold time of 3 min at each increment. The mass spectrometer was operated in the electron ionization mode at 70 eV and electron multiplier voltage at 1859 V. Other MS operating parameters were as follows: ion source temperature 230 °C, quadrupole temperature 150 °C, solvent delay 4 min and scan range 50–700 amu. The compounds were identified by direct comparison of the mass spectrum of the analyte at a particular retention time to that of reference standards found in the 2011 National Institute of Standards and Technology (NIST) library. The area percentage of each component was calculated by comparing its average peak area to the total area obtained. 

### 4.4. Ultra-High Performance Liquid Chromatography-MS/MS (UHPLC) Analysis of Phenolic Acids 

Phenolic acids present in aqueous (80%) methanol extracts of the different plant parts were identified and quantified using Ultra-high performance liquid chromatography-tandem mass spectrometry (UHPLC-MS/MS) as previously described [58]. The assay was done in triplicate and results presented as mean ± standard error. The mean values obtained were compared using one-way analysis of variance (ANOVA) and where statistical significance (*p* ≤ 0.05) existed, these values were further separated using the Duncan’s multiple range test. Bio-statistical analysis were done using the SPSS version 24.0 for Windows (IBM SPSS Inc., Chicago, IL, USA). 

### 4.5. Ames Test

The mutagenic properties of different extracts from the plant were evaluated using the Ames Salmonella/ -Microsome assay involving two *Salmonella typhimurium* tester strains, TA98 and TA102, in the absence of S9 metabolic activation [59,60]. One hundred microliters of sterile distilled water served as the negative control, while 2 µg/plate of 4NQO were used as the positive control. The assay was conducted twice, and results presented as mean ± standard error number of reverted colonies per plate. Plant samples that induced a 2-fold increase in the number of His^+^ revertants compared to the negative control were considered to be mutagenic. Additionally, samples that exhibited a dose-dependent increase in the number of His^+^ revertants were classified as mutagenic [26].

## 5. Conclusions

Natural products and their derivatives have, of late, attracted much attention as potential sources of drug leads that could be effective in combating microbial infections prevalent in humans. The present study revealed that *Protea caffra* is a potential source of antibacterial compounds effective against both drug-sensitive and -resistant bacterial strains. It is, however, pertinent that the antibacterial compounds in the plant be unequivocally identified and their mode of action elucidated. This is the first report on the pharmacologically properties of *Protea caffra* and based on the current findings, it is recommended that the other members of *Protea* species (>300) be explored for potential therapeutic properties. 

## Figures and Tables

**Table 1 plants-09-01331-t001:** Minimum inhibitory concentration values (MIC, mg/mL) of *Protea caffra* extracts screened against drug-sensitive and -resistant bacterial strains.

Minimum Inhibitory Concentration (MIC, mg/mL)													
	Methanol					Dichloromethane				Petroleum Ether			
Plant Part	*Ec*	*Ef*	*Kp*	*Sa*	*Sa* D	*Ec*	*Ef*	*Kp*	*Sa*	*Ec*	*Ef*	*Kp*	*Sa*
Bark	1.25	**0.63**	>2.5	**0.31**	>2.5	1.25	>2.5	2.5	1.25	1.25	>2.5	2.5	1.25
Flowers	1.25	1.25	1.25	2.5	2.5	1.25	**0.63**	1.25	**0.63**	1.25	**0.63**	1.25	**0.63**
Leaves	2.5	**0.63**	1.25	1.25	**0.63**	2.5	>2.5	1.25	2.5	2.5	>2.5	1.25	2.5
Seeds	**0.63**	1.25	**0.31**	1.25	1.25	**0.63**	**0.63**	**0.31**	1.25	**0.63**	**0.63**	**0.31**	1.25
Twigs	**0.63**	**0.31**	**0.63**	**0.31**	**0.63**	2.5	2.5	2.5	2.5	2.5	2.5	2.5	2.5
**Neomycin ^a^ (µg/mL)**	0.78	0.39	1.6	0.65	6.25								

*Ec* = *Escherichia coli*; *Ef* = *Enterococcus faecalis*; *Kp* = *Klebsiella pneumoniae*; *Sa* = *Staphylococcus aureus*, *Sa* D = Penicillin-resistant *S*. *aureus*. MIC values bold-written indicate noteworthy antibacterial activity. ^a^ = Positive control.

**Table 2 plants-09-01331-t002:** Minimum inhibitory concentration (MIC, mg/mL) values of fractions obtained from a methanol extract of *Protea caffra* twigs.

	Minimum Inhibitory Concentration (MIC, mg/mL)						
Fraction	*Ec*	*Ec* D	*Ef*	*Kp*	*Kp* D	*Sa*	*Sa* D
Acetone	>2.5	>2.5	>2.5	>2.5	>2.5	1.25	>2.5
Cold ethyl acetate	**0.6**	>2.5	**0.078**	**0.3**	>2.5	**0.15**	>2.5
Hot ethyl acetate	>2.5	>2.5	**0.3**	>2.5	>2.5	**0.6**	>2.5
MeOH	>2.5	>2.5	>2.5	> 2.5	>2.5	> 2.5	>2.5
**Neomycin ^a^ (µg/mL)**	0.78	6.4	0.39	1.6	8.3	0.65	6.3

^a^ = positive control, Ec D = Multidrug-resistant *Escherichia coli*, Ef = *Enterococcus faecalis*, Kp = *Klebsiella pneumoniae*, Kp D = Multidrug-resistant *Klebsiella pneumoniae*, Sa = *Staphylococcus aureus*, Sa D = Penicillin-resistant *Staphylococcus aureus*, MeOH = Methanol. *Values in bold are considered noteworthy antibacterial activity.

**Table 3 plants-09-01331-t003:** Gas Chromatograph-Mass Spectroscopy (GC-MS) data of phytocompounds putatively identified in different sub-fractions of methanolic extracts of *Protea caffra* twigs.

Sub-Fraction	Chemical Name	Retention Time	Area %	Similarity %	Molecular Formulae	Molecular Weight
**Cold ethyl acetate**	Polygalitol	13.35	34.76	95	C_6_H_12_O_5_	164
	Phenol, 4-(1,1,3,3-tetramethylbutyl)-	15.38	9.8	89	C_14_H_22_O	206
	Spiro-1-(cyclohex-2-ene)-2’-(5’-oxabicyclol)	14.94	8.2	77	C_14_H_22_O	206
	1-Adamantanecarboxylic acid, 2-propenyl	15.03	8.07	89	C_14_H_20_O_2_	220
	Carbamic acid, N-[1,1-bis(trifluoromethyl)ethyl	14.85	7.03	90	C_19_H_25_F_6_NO_2_	413
	Phenol, 2-methyl-4-(1,1,3,3-tetramethylbutyl)-	15.27	6.23	80	C_15_H_24_O	220
	Hexestrol, O-acetyl-	15.16	3.57	88	C_20_H_24_O_3_	312
	1-Heptacosanol	19.41	2.86	94	C_27_H_56_0	396
	1,2-Bis(p-acetoxyphenyl)ethanedione	14.74	2.1	77	C_18_H_14_0_6_	326
	1,3-Benzenediol, 4-propyl-	13.73	1.83	83	C_9_H_12_0_2_	152
	Phthalic acid, butyl tridecyl ester	18.53	1.74	75	C_22_H_28_O_4_	356
	1-Nonadecanol	15.89	1.60	93	C_19_H_40_O	284
	Phenol 2,4-bis (1,1-dimethylethyl)	12.84	1.16	94	C_14_H_22_O	206
**Hot ethyl acetate**	1-Heptacosanol	14.16	70.57	90	C_27_H_56_0	396
	1,3,5-Benzenetriol	13.55	15.42	95	C_6_H_6_0_3_	126
	Polygalitol	12.99	7.31	72	C_6_H_12_O_5_	164
	1,3-Benzenediol, 4-propyl-	13.73	3.93	70	C_9_H_12_O_2_	152
	ß-Glucopyranose, 1,6-anhydro-(levoglucosan)	11.60	2.78	70	C_6_H_10_O_5_	162
	1-Nonadecanol	15.89	1.6	93	C_19_H_40_O	284
**Acetone**	Oxalyl acid	3.23	51.12	93	C_2_C_I2_O_2_	126
	Polygalitol	13.22	48.88	95	C_6_H_12_O_5_	164

**Table 4 plants-09-01331-t004:** Quantity (µg/g DW) of hydroxybenzoic acids in 80% methanol *Protea caffra* extracts. Values represent mean ± standard error, *n* = 3.

Hydroxybenzoic Acids (µg/g DW)							
Plant Part	Catechin Acid	Gallic Acid	*p*-Hydroxybenzoic Acid	*p*-Protocatechuic Acid	Salicylic Acid	Syringic Acid	Vanillic Acid
Bark	6.83 ± 1.4 ^c^	0.3 ± 0.3 ^b^	1.95 ± 0.1 ^d^	88.1 ± 4.4 ^b^	0.1 ± 0 ^e^	2.4 ± 0.4 ^a^	4.7 ± 0.2 ^c^
Leaves	17.98 ± 0.51 ^a^	1.50 ± 0.03 ^a^	374.55 ± 9.14 ^a^	184.35 ± 4.44 ^a^	5.07 ± 0.08 ^b^	1.22 ± 0.02 ^c^	23.35 ± 0.13 ^a^
Seeds	<LOD	0.69 ± 0.23 ^b^	23.74 ± 0.37 ^c^	44.22 ± 1.32 ^d^	1.55 ± 0.04 ^d^	0.67 ± 0.03 ^d^	4.8 ± 9 0.2^c^
Twigs	13.02 ± 2.03 ^b^	0.42 ± 0.02 ^b^	4.94 ± 0.36 ^d^	17.65 ± 0.76 ^e^	1.85 ± 0.03^c^	1.28 ± 0.3 ^c^	6.66 ± 0.09 ^c^
Flowers	4.08 ± 1.3 ^d^	1.63 ± 0.04 ^a^	156.87 ± 5.03 ^b^	50.35 ± 1.57 ^c^	7.73 ± 0.36 ^a^	1.58 ± 0.07 ^b,c^	10.86 ± 0.7 ^b^

In each column, values with different letters are significantly different (*p* ≤ 0.05) as separated by Duncan’s Multiple Range Test.

**Table 5 plants-09-01331-t005:** Quantity (µg/g DW) of hydroxycinnamic acids in 80% methanol *Protea caffra* extracts. Values represent mean ± standard error, *n* = 3.

	Hydroxycinnamic Acids (µg/g DW)				
Plant Part	Caffeic Acid	Chlorogenic Acid	*p*-Coumaric Acid	Ferulic Acid	Sinapic Acid
Bark	5.69 ± 0.22 ^e^	0.56 ± 0.03 ^d^	1.69 ± 0.06 ^d^	14.12 ± 0.39 ^a^	0.2 ± 0 ^b^
Leaves	266.37 ± 1.46 ^a^	11.67 ± 0.28 ^b^	21.22 ± 0.54 ^a^	2.68 ± 0.18 ^e^	0.08 ± 0 ^c^
Seeds	29.09 ± 0.77 ^c^	3.34 ± 0.16 ^c^	10.03 ± 0.4 ^b^	4.89 ± 0.2 ^d^	0.09 ± 0.01 ^c^
Twigs	9.36 ± 0.39 ^d^	29.91 ± 1.21 ^a^	5.3 ± 0.26 ^c^	5.68 ± 0.26 ^c^	0.23 ± 0.13 ^a^
Flowers	39.36 ± 1.39 ^b^	4.82 ± 0.2 ^c^	10.77 ± 0.62 ^b^	6.92 ± 0.13 ^b^	0.24 ± 0.03 ^a^

In each column, values with different letters are significantly different (*p* ≤ 0.05) as separated by Duncan’s Multiple Range Test.

**Table 6 plants-09-01331-t006:** Number of His^+^ revertants in *Salmonella typhimurium* strains TA98 and TA102 produced by *Protea caffra* organic solvent extracts.

	Number of His^+^ Revertants/Plate (mg/mL)						
		TA98			TA102		
**Plant organ**	**Solvent**	**5**	**0.5**	**0.05**	**5**	**0.5**	**0.05**
Bark	MeOH	9.7 ± 2.3	8.3 ± 3.2	10.7 ± 1.5	248.0 ± 20.1	174.7 ± 11.7	177.3 ± 15.5
	PE	6.4 ± 2.1	19.8 ± 4.1	15.8 ± 6.1	213.5 ± 15.9	172.5 ± 26.2	245.6 ± 21.1
Flowers	PE	13.6 ± 5.2	8.8 ± 2.6	12.6 ± 3.2	228.0 ± 12.1	145.3 ± 22.5	207.0 ± 12.8
Seeds	DCM	18.0 ± 4.6	7.0 ± 3.6	8.7 ± 4.7	269.3 ± 22.0	253.3 ± 28.3	264.0 ± 5.6
	MeOH	9.7 ± 2.1	10.0 ± 2.6	10.7 ± 2.1	225.3 ± 21.1	278.7 ± 8.5	260.0 ± 2.6
	PE	8.7 ± 3.2	11.6 ± 1.9	21.9 ± 7.6	269.0 ± 33.6	276.0 ± 18.2	208.0 ± 25.6
Twigs	MeOH	22.0 ± 6.2	29.0 ± 8.2	17.3 ± 6.7	192.0 ± 6.2	235.0 ± 22.5	284.3 ± 27.4
Leaves	MeOH	16.7 ± 1.5	15.7 ± 11.0	27.0 ± 11.4	278.0 ± 30.2	109.7 ± 22.0	281.0 ± 43.9
	PE	7.2 ± 2.3	22.8 ± 7.6	19.5 ± 4.9	233.6 ± 16.5	246.0 ± 17.7	217.6 ± 14.7
	Water (-ve control)		19.1 ± 8.4			145.2 ± 17	
4-nitroquinoline-oxide (+ve control)			191.9 ± 17.3			296.7 ± 20.6	

The data presented are the mean ± standard error of six plates from two separate experiments each performed in triplicate. DCM = Dichloromethane, MeOH = Methanol, PE = Petroleum ether, -ve= negative. + ve = positive control.

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
