# Peer review of "Antibacterial, Mutagenic Properties and Chemical Characterisation of Sugar Bush (Protea caffra Meisn.): A South African Native Shrub Species"

_plants, 2020, doi:10.3390/plants9101331_

Round 1

Reviewer 1 Report

Medicinal plants and natural products have attracted much attention as potential sources of pharmaceuticals.
The manuscripts concers the up-to-date question of the use of natural compounds in human therapy. The research revealed the potential of Protea caffra as an antibacterial agent against both drug-sensitive and -resistant bacterial strains and it is a promissing starting point for futher studies on the medicinal use of Protea caffra.
I believe the manuscript could be published after a minor revision.

Personally, I would modify/enrich the introduction and disscusion:
* line 40: the Authors report that "In 2010 alone, over 1.7 billion cases of childhood diarrhoea were recorded worldwide, 40 30% of which led to death" - are there any newer data on diarrhoea problem?
* line 43: "Globally, medicinal plants are used to manage diarrheal symptoms" - please provide the examples; in what form (plant infusion, extracts, their derivatives)?
* disscusion - to ilustrate the antimicrobial potential of Protea caffra in comparison with other medicinal plants, please provide the data on e.g. minimum inhibitory concentration values of extracts from different plant species used in therapy of diarrhoea.

Reviewer 2 Report

The manuscript describes the antibacterial, mutagenic properties and chemical characterization of Protea caffra, a species of shrubs native to South Africa. Since this plant is traditionally used in South Africa by herbalists to manage diarrheal symptoms, the authors have attempted to establish the scientific basis of this use. Several medicinal plants are traditionally used to treat different diseases without any phytochemical and toxicological knowledge. Therefore, the study and characterization of these valuable plants have great scientific merit and is a poorly explored scientific field. It is a good study. However, the study is too unspecific.

The introduction is not really sufficient, how do herbalists use the plant? which parts of the plant are usually used? Are they the ones in the study? They use it in decoction, infusion, maceration….? because I don't think they use organic solvents.

It is very good to used different organic solvents. But, in my opinion, to correlate the results obtained with what is traditionally found, it could have been very interesting to analyze the sample as it is traditionally prepared and to compare the results with those obtained by solvent extraction… the phytochemical and toxicological data differ drastically depending on the extraction method…so, it is not known if the herbalist’s preparation contains all the compounds mentioned. The results cannot be correlated to the management of diarrheal symptoms and do not really shed light on the issues raised in line 57-58.

Compound identification is very basic, the authors could use the authentic standards to confirm the presence of potentially interesting or abundant compounds…

Are 1-Adamantyl, levoglucosan, as well as most of the detected compounds also generally found in plants?

I do not really understand the negative control of the antibacterial test. Why did the authors use sterile distilled water and 2% DSMO? Why not sterile distilled water and 2% of respective solvents used for the extraction?

Line 156-157: Chlamydia infection is not a bacterial infection, but rather a fungal infection.

Round 2

Reviewer 2 Report

Dear authors,

The reviewer thanks the authors for the good clarification of the raised concerns and the improvement of the manuscript.

However,  data on the traditional preparation of the extracts are missing. In my opinion, the chemical composition of this preparation should be part of the word.  The authors tested the extracts as traditionally prepared and found no effect this is not discussed in the manuscript. The composition of this preparation is unknown and may differ from the solvent extracted one. Is it the concentrations/compositions of the active compounds which make the difference in the observed effects between the two different preparations?  

As presented now, the study revealed clearly that some parts of the plant which are traditionally used as a remedy, have interesting bioactive compounds. But, the results cannot be correlated to the management of the diseases and do not really shed light on the traditional use of the plant.

Round 3

Reviewer 2 Report

Dear authors,

The reviewer thanks the authors for the good clarification of the raised concerns and the improvement of the manuscript. However, data on the traditional preparation of the extracts are missing. In my opinion, the chemical composition of this preparation should be part of the word. The authors tested the extracts as traditionally prepared and found no effect this is not discussed in the manuscript. The composition of this preparation is unknown and may differ from the solvent extracted one. Is it the concentrations/compositions of the active compounds which make the difference in the observed effects between the two different preparations? As presented now, the study revealed clearly that some parts of the plant which are traditionally used as a remedy, have interesting bioactive compounds. But, the results cannot be correlated to the management of the diseases and do not really shed light on the traditional use of the plant.

 RESPONSE: You made a valid argument, however, it is important to note that in the present investigation, we did not necessary follow the traditional method of preparing the extract. Given that water is most common and easily available solvent for traditionally herbal extract in folk medicine, we included the water extract at the initial screening stage in order to mimic the traditional preparation of the extract. However, this demonstrated very weak antibacterial activity. We have alluded to this in the manuscript (please see lines 76-77). REVIEWER COMMENT: Yes and these were also important results that I think are interesting for the reader!

 It is common practice that the findings from the preliminary screening strongly influenced the subsequent experiments especially from a bioactivity guided approach. In our case, the result (Minimum inhibitory concentration of above 2.5 mg/ml) suggest that the bioactive compounds that is response for antibacterial effect is relatively low in the water extracts. Based on polarity index, water is known to extract a wide range of compounds including sugars. To some extent, most water extracts do not exhibit biological activity and this has been widely reported (Eloff, 2019; Jager, 2003). REVIEWER COMMENT: Yes I agree completely but this is what I am pointing out… you should discuss these results as above in the manuscript instead to exclude those results (may be you can add them in the supplement if you do not want them in the manuscript)…And it is also known that different solvents, based on their polarities, have a limitation to what phytocompounds they can extract.

The water extract in this study was not further analyzed solely due to the lack of antibacterial activity. REVIEWER COMMENT: This could be due to the fact that “the bioactive compounds that are responsible for antibacterial effect are relatively low in the water extracts” as you mentioned above… Based on the objective of our study, the next logical step was to follow-up on the most active extracts. This approach is well justified and in line with common practice in the field. Please see Table 1 of the paper on ‘Best practice in research – Overcoming common challenges in phytopharmacological research’ by Heinrich et al (2020). As established by ‘Assessment of the ‘translatability’ of traditional medical concepts and uses, even though the translation of traditional medicine concepts in Western medicine will often be problematic’. In fact, the approach is never to validate traditional medicine using western approach but to provide evidence where necessary. REVIEWER COMMENT:  I understand and I still tink to provide evidence, the characterization of the traditional preparation is also very important.

Even if we want to profile the water extract of P. caffra (just for comparison, it is not feasible at this stage. The study was done a while ago and plant materials have been exhausted. In order to answer the question, it will mean doing ALL the work again as that is the only way to have a valid response to the query. REVIEWER COMMENT: In my opinion,  if you can't measure, the least you can do is to discuss the results obtained with this extract on microorganisms as I already mention above…

In conclusion, the reviewer agreed that ‘the study revealed clearly that some parts of the plant which are traditionally used as a remedy, have interesting bioactive compounds’. In our own opinion, the absence of the profile of the water extracts (used in traditional medicine) do not nullify our findings (which is submitted as ‘a Brief communication’ to this journal). REVIEWER COMMENT: Yes you obtained very good and interesting results, but it could have been more interesting and beneficial for the readers if you could have shed more light on the issues raised in line 57-58 by flowing the water extract characterization. Nevertheless, I think you should try to discuss the results obtained from the effect of this extract accordingly.

Thank you for the kind consideration.
